# Structure, gating, and pharmacology of human Ca$_V$3.3 channel

Lingli He[1,2,3,8], Zhuoya Yu[1,2,3,8], Ze Geng[4,5,8], Zhuo Huang [4,5,8], Changjiang Zhang[2,3], Yanli Dong [1,2], Yiwei Gao [1,2,3], Yuhang Wang[1,2,3], Qihao Chen[1,2,3], Le Sun[6], Xinyue Ma[4,5], Bo Huang [7], Xiaoqun Wang [2,3] & Yan Zhao [1,2,3 ✉]

The low-voltage activated T-type calcium channels regulate cellular excitability and oscillatory behavior of resting membrane potential which trigger many physiological events and have been implicated with many diseases. Here, we determine structures of the human T-type Ca$_V$3.3 channel, in the absence and presence of antihypertensive drug mibefradil, antispasmodic drug otilonium bromide and antipsychotic drug pimozide. Ca$_V$3.3 contains a long bended S6 helix from domain III, with a positive charged region protruding into the cytosol, which is critical for T-type Ca$_V$ channel activation at low voltage. The drug-bound structures clearly illustrate how these structurally different compounds bind to the same central cavity inside the Ca$_V$3.3 channel, but are mediated by significantly distinct interactions between drugs and their surrounding residues. Phospholipid molecules penetrate into the central cavity in various extent to shape the binding pocket and play important roles in stabilizing the inhibitor. These structures elucidate mechanisms of channel gating, drug recognition, and actions, thus pointing the way to developing potent and subtype-specific drug for therapeutic treatments of related disorders.

[1] National Laboratory of Biomacromolecules, CAS Center for Excellence in Biomacromolecules, Institute of Biophysics, Chinese Academy of Sciences, Beijing 100101, China. [2] State Key Laboratory of Brain and Cognitive Science, Institute of Biophysics, Chinese Academy of Sciences, 15 Datun Road, Beijing 100101, China. [3] College of Life Sciences, University of Chinese Academy of Sciences, Beijing 100049, China. [4] State Key Laboratory of Natural and Biomimetic Drugs, Department of Molecular and Cellular Pharmacology, School of Pharmaceutical Sciences, Peking University Health Science Center, Beijing 100191, China. [5] IDG/McGovern Institute for Brain Research, Peking University, Beijing 100871, China. [6] Beijing Institute of Brain Disorders, Capital Medical University, Beijing 100069, China. [7] StoneWise Ltd., 1708, Block B, No.19 Zhongguancun Street, Haidian District, Beijing, China. [8] These authors contributed equally: Lingli He, Zhuoya Yu, Ze Geng, Zhuo Huang. ✉email: zhaoy@ibp.ac.cn

Voltage-gated calcium channels ($Ca_V$ channels) serve as a crucial transducer converting depolarization of membrane potential to local intracellular calcium signals, and thus initiate many physiological events, such as neuronal firing, pacemaker rhythms, neurotransmitter release as well as smooth muscle contraction[1–4]. $Ca_V$ channels can be traditionally classified into high-voltage activated (HVA) calcium channels (L, N, P/Q, and R-types), and low-voltage activated calcium channels such as T-type calcium channels[5–7]. Each $Ca_V$ subtype has distinct tissue-specific localization, electrophysiological properties, and pharmacological profiles. In particular, T-type $Ca_V$ channels, also termed as $Ca_V3$ channels, are widely expressed throughout the nervous, neuroendocrine, and cardiovascular systems[3,8–11]. Three T-type calcium channel isoforms, including α1G ($Ca_V3.1$), α1H ($Ca_V3.2$), and α1I ($Ca_V3.3$), have been identified in mammals and well characterized[12–14], and they display distinct kinetic properties concerning their rates and voltage dependency of activation and inactivation[12,15,16]. Among them, $Ca_V3.3$ is featured by slow activation and inactivation, suggesting that it plays a role in sustained firing[14].

Cumulative evidence shows that T-type calcium channels are important pharmacological targets in pathophysiological processes, including cardiac arrhythmia, hypertension, and neurological disorders[17,18]. Development of highly selective antagonists of T-type $Ca_V$ channels holds promise for therapeutic intervention in particular pathologies[19]. For instance, mibefradil (MIB) is the first antagonist that blocks T-type calcium channels and was initially launched on the market as a treatment for hypertension[20]. Some other clinically used drugs, such as antispasmodic drug otilonium bromide (OB) and antipsychotic drug pimozide (PMZ), are also able to potentially reduce the activity of $Ca_V3$ channels, and such effects may significantly contribute to their therapeutic efficacy[21–23].

In recent years, structures of different subtypes of $Ca_V$ channels have been reported in their apo state or in complex with different modulators, providing rich structural insights of the gating and modulation mechanisms of $Ca_V$ channels[24–31]. In particular, structure of the human $Ca_V3.1$ channel was resolved. However, the structural basis of modulation mechanism(s) of the current clinically used drugs, as mentioned above, remains elusive, yet such information should be of great importance for the deployment of next-generation therapeutic agents specifically targeting T-type $Ca_V$ channels. Moreover, despite the structure of the $Ca_V3.1$ has been determined, the mechanism for activating T-type $Ca_V$ channels at low voltages is still elusive.

In this work, we express and purify the human $Ca_V3.3$ channel in HEK293 cells. The construct is modified partially based on its splicing sites, which do not alter channel properties. We use the cryo-EM method and determine the $Ca_V3.3$ channel structures in its apo state, MIB-bound state, OB-bound state, and PMZ-bound state. These structures illustrate channel gating mechanism as well as how these drug molecules modulate the channel activity.

## Results and discussion

**Architecture of the human $CaV3.3$ channel**. To investigate modulation mechanism(s) of variety of channel blockers on T-type $Ca_V$ channels, we cloned and expressed the human $Ca_V3.3$ channel in HEK293 cells. To improve the expression level and homogeneity of the protein sample, the N-terminal region (residues 1−42) and C-terminal region (residues 2065−2223) were truncated, and an incidental point mutation Glu523-to-Gln (E523Q) was included. Whole-cell patch-clamp experiment confirmed that these modifications on the $Ca_V3.3$ construct resulted in similar voltage dependency of the activation and steady-state inactivation compared to full-length wild-type $Ca_V3.3$ (Fig. 1 and Supplementary Fig. 1). We named this construct as $Ca_V3.3^{EM}$ (EM stands for

electron microscopy), which was used for further cryo-EM study and electrophysiological experiments. We purified the $Ca_V3.3^{EM}$ protein and collected the cryo-EM data in the absence ($Ca_V3.3^{apo}$) or in the presence of distinct ligands, including Mibefradil ($Ca_V3.3^{MIB}$), Otilonium bromide ($Ca_V3.3^{OB}$) and Pimozide ($Ca_V3.3^{PMZ}$). The cryo-EM maps of $Ca_V3.3^{apo}$, $Ca_V3.3^{MIB}$, $Ca_V3.3^{PMZ}$, and $Ca_V3.3^{OB}$ complexes are determined at 3.3, 3.9, 3.6, and 3.6 Å, respectively (Supplementary Figs. 2, 3 and Supplementary Table 1), and they are rich in structural features, including densities for side chains, N-glycans, and associated lipid molecules. More importantly, the densities of distinct ligands were also well resolved and allowed us to unambiguously build atomic models of these $Ca_V3.3$ complexes.

A total of 1135 residues were built into the final model. The extracellular loops (ECLs), voltage-sensing domains (VSDs), and pore domain are well resolved (Fig. 1b and Supplementary Fig. 4). The missing parts include residues from linkers between domains, N-terminal or C-terminal loops, presumably due to conformational flexibility. The $Ca_V3.3$ channel consists of four repeated transmembrane domains, $D_I−D_{IV}$. Each domain comprises six transmembrane helices (S1−S6). Among them, the S1−S4 helices constitute the VSD; the S5 and S6 helices from all four domains form the ion-conducting pore; and the channel is formed in a domain-swapped fashion. Two re-entrant short helices, P1 and P2, connecting S5 and S6 of each domain, contribute to the selectivity filter (SF). The selectivity filter ring contains four acidic residues, E357, E821, D1380, and D1678 (EEDD motif), one from each domain, and together they produce a strong negative charged area to attract cations and determines the selection specificity for $Ca^{2+}$ ions[32,33]. We identified a strong density inside the selectivity filter and close to the EEDD motif (Fig. 1c). We speculated that it represents a calcium ion, consistent with the previously observations in $Ca_V$ channels structures[27,29]. In addition, several phospholipids intrude into the central cavity from the fenestration sites around domain interfaces (Supplementary Fig. 5b). The pore-lining helices (S6s) converged in the intracellular ends to form the intracellular gate (Fig. 1c). The pore profile calculated by the program HOLE[34] indicated that this gate is closed in the apo state structure.

**Gating mechanism of T-type CaV channel at low voltage**. The structure of the $Ca_V3.3^{apo}$ was compared with the $Ca_V3.1$ structure (PDB ID: 6KZO)[29], giving rise to an RMSD of ~1.4 Å for 942 Cα-pairs. The overall structure, including VSD, SF, and ECLs, is fairly superimposable (Fig. 2a and Supplementary Figs. 5a–c). Nevertheless, one obvious structural discrepancy is identified in the S6 helix from domain II ($S6_{II}$) (Fig. 2b). This helix in $Ca_V3.3^{apo}$ is featured by a π-bulge at Y851, but assumes an α-helical conformation in $Ca_V3.1^{apo}$. Moreover, a two-branched phospholipid molecule was also resolved previously in the $Ca_V3.1$ structure; nevertheless, it enters the central cavity from the $D_{II}$-$D_{III}$ fenestration site in $Ca_V3.1^{apo}$, rather than the $D_{III}$–$D_{IV}$ fenestration site as observed in $Ca_V3.3^{apo}$ (Supplementary Fig. 5b, c). We found that in the $Ca_V3.3^{apo}$ structure, residue Y851 occludes the $D_I$-$D_{II}$ fenestration site due to the π-bulge on $S6_{II}$ and thus prohibits insertion of a phospholipid (Fig. 2b). Furthermore, the $S6_{III}$ helix in $Ca_V3.3^{apo}$ appears to be longer by six turns and more bended, thus extends into the cytosol and directly contacts with $VSD_{IV}$. In contrast, in the $Ca_V3.1^{apo}$ structure, the cytoplasmic part of the equivalent S6 helix was not determined (Fig. 2a), although the amino acid sequences are conserved (Fig. 2c). We noticed that the loop between $S6_{III}$ helix and $VSD_{IV}$ (loop$^{III-IV}$) in $Ca_V3.1$ is longer than in $Ca_V3.3$. We speculate this long loop$^{III-IV}$ accounts for high flexibility of $S6_{III}$ in $Ca_V3.1$. Moreover, the loop$^{III-IV}$ of $Ca_V3.1$ can be alternatively spliced into different isoforms, which displays distinct gating properties[35,36], probably by affecting conformation of the $S6_{III}$ helix.

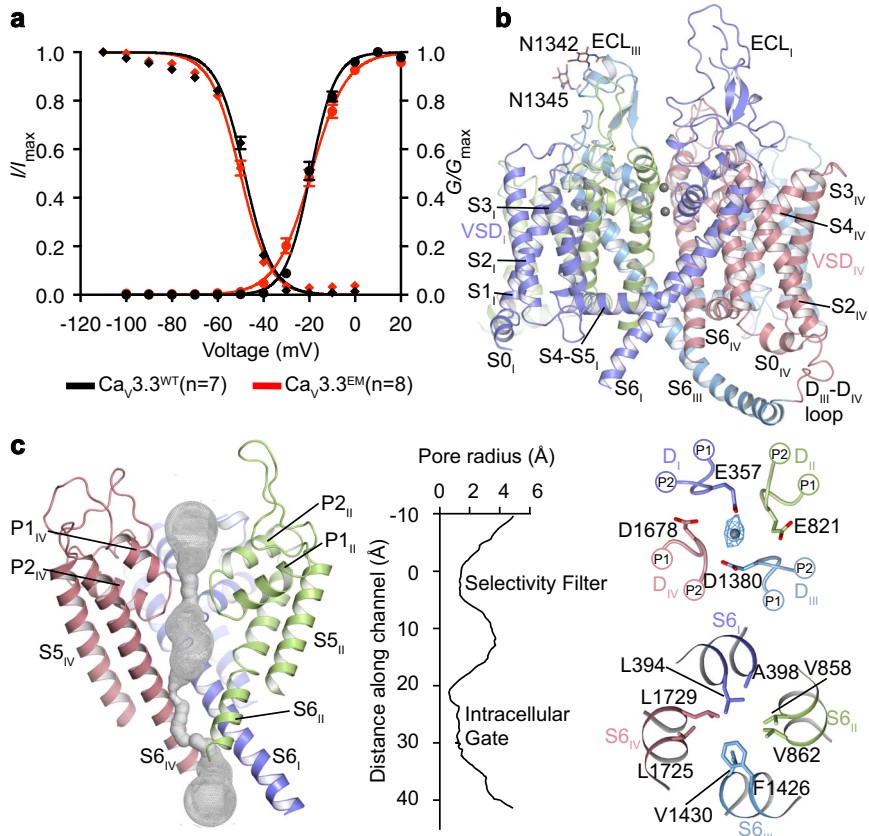

**Fig. 1 Architecture of the Ca$_V$3.3 channel. a** The activation (right) and inactivation (left) properties of the truncated Ca$_V$3.3$^{EM}$ and full-length wild type (WT) Ca$_V$3.3$^{WT}$. Normalized conductance ($G/G_{max}$) for the activation properties is applied by a held at −100 mV and then a series of 200 ms voltage steps from −100 mV to +20 mV in 10 mV increments. The ion current ($I/I_{max}$) for inactivation (left) properties are assessed with a 3.6 s holding voltages ranging from −110 mV to 0 mV (10 mV increments) followed by a 150 ms test pulse at −20 mV. N cells were analyzed (WT, $n = 7$; EM, $n = 8$). Data are presented as mean values +/− SEM. Source data are provided as a Source Data file. **b** The overall structure of the Ca$_V$3.3$^{apo}$. The domains of Ca$_V$3.3 are colored as D$_I$ in purple, D$_{II}$ in green, D$_{III}$ in blue, and D$_{IV}$ in salmon. A cation ion in the selectivity filter vestibule was presented as a gray sphere. **c** The ion permeation path in the pore domain. The selectivity filter and S5-S6 helices are shown in cartoon and viewed in parallel to the membrane plane. The ion-conducting pathway was calculated by the program HOLE and illustrated by gray dots in the left panel. The selectivity filter ring of four negatively charged residues from the four domains is shown in sticks in the upper, and the intracellular gate formed by four S6 helix viewed from extracellular side was shown in the lower of the right panel, respectively.

The Ca$_V$3.3 structure is further compared with the structure of a high-voltage activated Ca$_V$2.2 channel[31] (PDB ID: 7VFS), giving rise to an RMSD of ~2.8 Å for 930 Cα-pairs. Some discernible structural differences are observed occurring at the extracellular loops, which would cause incompatibility of Ca$_V$3.3 with the Ca$_V$2.2 accessory subunit α2δ, as discussed in the previous report[29]. Most importantly, the S6$_{III}$ helix in the Ca$_V$2.2 structure is much shorter than that in Ca$_V$3.3 as well (Fig. 2d). However, unlike the high flexibility of S6$_{III}$ resulting in shorter S6 helix in the Ca$_V$3.1 structure, the shorter S6$_{III}$ helix in Ca$_V$2.2 is due to the C-terminal cytoplasmic extension of S6$_{III}$ in Ca$_V$3.3 is not conserved in Ca$_V$2.2 as well as other HVA Ca$_V$ channels (Fig. 2c). According to sequence alignment, this long bended S6$_{III}$ helix determined in the Ca$_V$3.3 is conserved in all low voltage-activated T-type Ca$_V$ channels, but absent in HVA Ca$_V$ channels (Fig. 2c). Thus, we defined two regions of the S6$_{III}$ helix, including transmembrane pore-lining region (S6$^{TM}$) and cytoplasmic positive charged region (S6$^{Cyto}$) (Fig. 2d). Taking a closer look, the S6$^{Cyto}$ region is rich in positive charged residues ($^{1445}$EAR-RREEKRLRRLEKKRRK$^{1463}$). To explore functional roles of the extra-positively charged S6$^{Cyto}$ region, we mutated all the arginine and lysine to glutamine (Ca$_V$3.3$^{12Q}$) and carried out an electrophysiological experiment. This mutation shifted the voltage dependency of the activation curve toward more depolarized

membrane potential; the midpoint of the curve ($V_{1/2}$) is ~−20 mV for WT Ca$_V$3.3, and ~−5 mV for Ca$_V$3.3$^{12Q}$, while the change of voltage dependency of steady-state inactivation is subtle (Fig. 2e). These results demonstrated that the positive charged S6$^{Cyto}$ portion senses the membrane potential and thus contributes partially to low activation threshold of the T-type Ca$_V$ channels compared with the HVA Ca$_V$1 and Ca$_V$2 channels. We further speculate that the rigidity between S6$^{TM}$ and S6$^{Cyto}$ is critical for S6$^{Cyto}$ to confer additional voltage sensitivity to the channel. To validate this hypothesis, we mutated the "$^{1442}$EAE$^{1444}$" to "GGG" (Ca$_V$3.3$^{3G}$), which is likely to break the whole S6$_{III}$ helix into two helical segments and to induce more flexibility into the S6$^{Cyto}$ relative to the S6$^{TM}$ portion. Electrophysiological experiment indicated that the Ca$_V$3.3$^{3G}$ mutant displays similar changes in the gating properties as Ca$_V$3.3$^{12Q}$ (Fig. 2e), implying that the triple glycine mutation decouples the voltage sensing of S6$^{Cyto}$ from the channel activation. Considering the fact that the number of the gating charges of each VSD is identical between Ca$_V$3.3 and Ca$_V$2.2 and two channels also share similar hydrophilic cavities (Supplementary Fig. 5d), we speculate that the low voltage-gating property of T-type Ca$_V$ channels is partly due to the positively charged S6$^{Cyto}$. This cytosolic helix extension provides additional voltage sensitivity to the channel gate directly through the S6$_{III}$ helix, and its rigidity is essential for the S6$_{III}$ helix to exert modulation effect

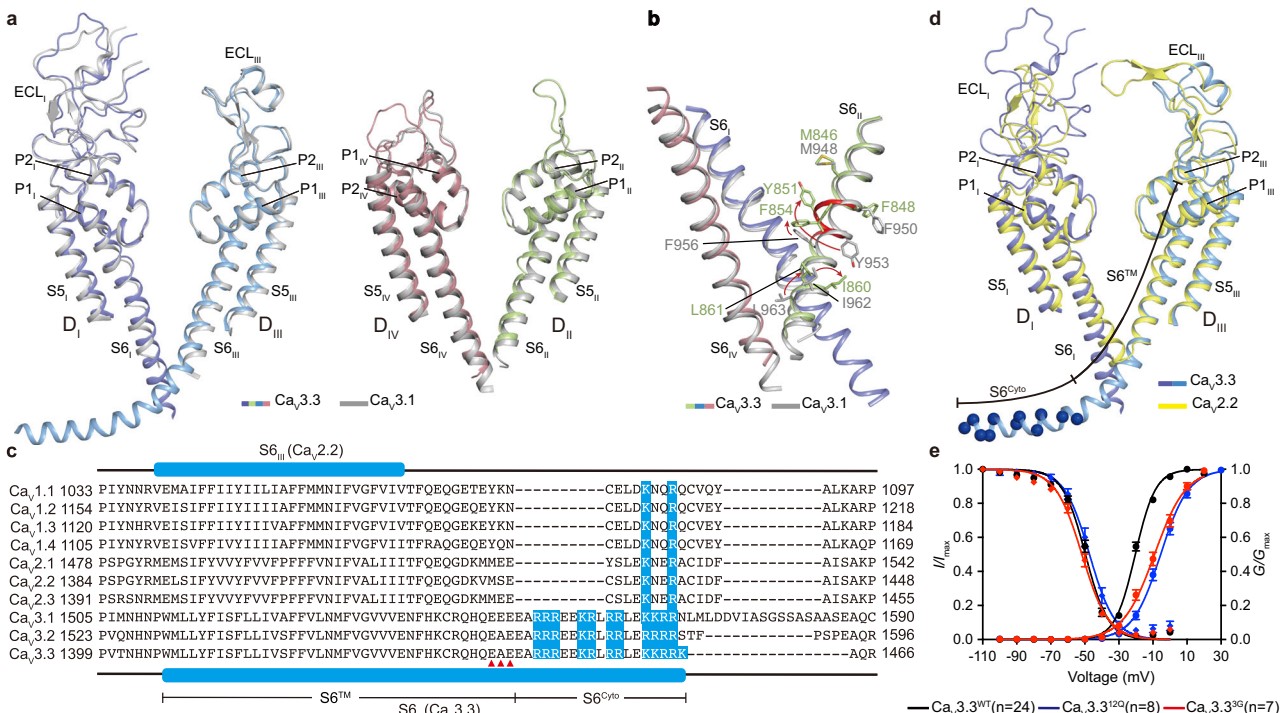

**Fig. 2 Positive charged S6$^{Cyto}$ region is critical for the channel gating. a** The pore domain segments in the D$_I$/D$_{III}$ (left panel) and D$_{II}$/D$_{IV}$ (right panel) of the Ca$_V$3.3 (colored), superimposed with the corresponding region of Ca$_V$3.1 structure (gray). **b** Zoom-in view highlights the structural comparison S6$_{II}$ helix between Ca$_V$3.3$^{apo}$ and Ca$_V$3.1 (PDB ID: 6KZO) (gray). The shifts of the side chain of residue on S6$_{II}$ helix between Ca$_V$3.3$^{apo}$ and Ca$_V$3.1 undergoing axial rotation are indicated by red arrows. **c** Sequence alignment of S6$_{III}$ helix of all human Ca$_V$ members, numbered according to full-length subunits. The secondary structure of Ca$_V$2.2 and Ca$_V$3.3 marked above and below the sequence alignment, respectively. The dashes represent gaps. The positively charged residues on S6$_{III}$ are shaded in blue. Residues involved in Ca$_V$3.3$^{3G}$ are marked by red triangle. **d** The pore domain segments in the D$_I$/D$_{III}$ of the Ca$_V$3.3$^{apo}$ (colorful) and Ca$_V$2.2 (PDB ID: 7VFS) (yellow) are superimposed. The S6$^{Cyto}$ and S6$^{TM}$ regions are indicated. The positively charged residues on the S6$^{Cyto}$ region are shown as blue spheres. e. Normalized conductance-voltage (G/V) and current-voltage(I/V) relationship for the Ca$_V$3.3$^{WT}$ construct (black) and mutant Ca$_V$3.3$^{12Q}$ (blue), triple-mutant Ca$_V$3.3$^{3G}$ (red). n represents the number of repeated measurements. N cells were analyzed (WT, $n = 24$; Ca$_V$3.3$^{12Q}$, $n = 8$; Ca$_V$3.3$^{3G}$, $n = 7$). Data are presented as mean values ± SEM. Source data are provided as a Source Data file.

on gating properties. Previous reports have shown that the opening of the ion-conductive pore of T-type calcium channels does not require activation of all four VSDs and the gating brake located within the I–II loop plays an important role in regulating the channel opening[37–40]. We speculate this extended and positively charged S6$_{III}$ helix may cooperate with VSDs and gating brake to modulate the gating mechanism of T-type channel. However, more studies are required to fully understand this potential synergistic regulation mechanism.

**Mechanism of mibefradil antagonism.** Mibefradil (MIB) is a benzimidazoyl-substituted tetraline derivative that act as a higher affinity blocker for T-type Ca$_V$ channels than for HVA L-type Ca$_V$ channels[20,41,42]. Since T-type Ca$_V$ channels participate in cardiac pacemaker activity, mibefradil was launched on the market as an antihypertensive and antianginal agent[43]. Due to harmful interactions with other drugs, mibefradil was withdrawn from the market[44]. However, it is still desirable to understand molecular details how mibefradil blocks T-type Ca$_V$ channels with higher affinity. We determined the structure of the Ca$_V$3.3$^{MIB}$ complex at 3.9-Å resolution (Supplementary Fig. 3 and Supplementary Table 1). The MIB molecule adopts a 'Z' shape, is located in the central cavity of the pore domain, and fits well to the density (Fig. 3a–c). The benzimidazole group is flanked by both S6$_{II}$ and S6$_{III}$ helices, forming close hydrophobic contacts with surrounding residues (F854, L818, and L1415) (Fig. 3d, e). It is also engaged in hydrogen bonds with N850 and L818 on C-terminal of the P1 helix (Fig. 3e). The hydrophobic flunaphthalene and isopropyl groups point downward, sit above the

intracellular gate, and are involved in extensive hydrophobic interactions with residues F1426, L1721, and I1722 at the crossing site between S6$_{II}$ and S6$_{III}$ (Fig. 3d). Moreover, the methoxyacetate group extends along S6$_{III}$ and is coordinated by forming hydrogen bonds with K1379 and S1419 (Fig. 3e). Furthermore, the abovementioned two-tailed lipid, which penetrates into the cavity, lays above the low part of the mibefradil molecule and thus stabilizes the latter (Fig. 3d). In particular, the phosphate group of the lipid molecule is slightly displaced towards MIB relative to that in Ca$_V$3.3$^{apo}$ and consequently is positioned right underneath the selectivity filter. Such a movement is likely to hinder influx of Ca$^{2+}$ through the selectivity filter. MIB has also been characterized to block HVA L-type Ca$_V$ channels (e.g., Ca$_V$1.1 and Ca$_V$1.2), yet with much lower efficacy (~10- to 15-fold lower)[20,41,42]. Superimposition of Ca$_V$3.3$^{MIB}$ complex with HVA L-type Ca$_V$1.1 channel (PBD ID: 5GJV) resulted in an RMSD of ~2.5 Å for 920 Cα-pairs and no obvious steric clash is observed between the MIB and residues from Ca$_V$1.1. However, F854 and K1379 are replaced by L652 and F1013 in Ca$_V$1.1, respectively, and these substitutions may account for reduced binding affinity of MIB to Ca$_V$1.1 (Fig. 3f).

**Mechanism of otilonium bromide antagonism.** OB is an antispasmodic drug and is used worldwide for the treatment of irritable bowel syndrome[45,46]. OB inhibits gastrointestinal motility through several mechanisms, for example by blocking L-type and T-type Ca$_V$ channels, which are expressed in gastrointestinal smooth muscle cells and play pivotal roles in regulating contractility[47]. OB consists of N-octyloxy, benzoyl, benzocaine,

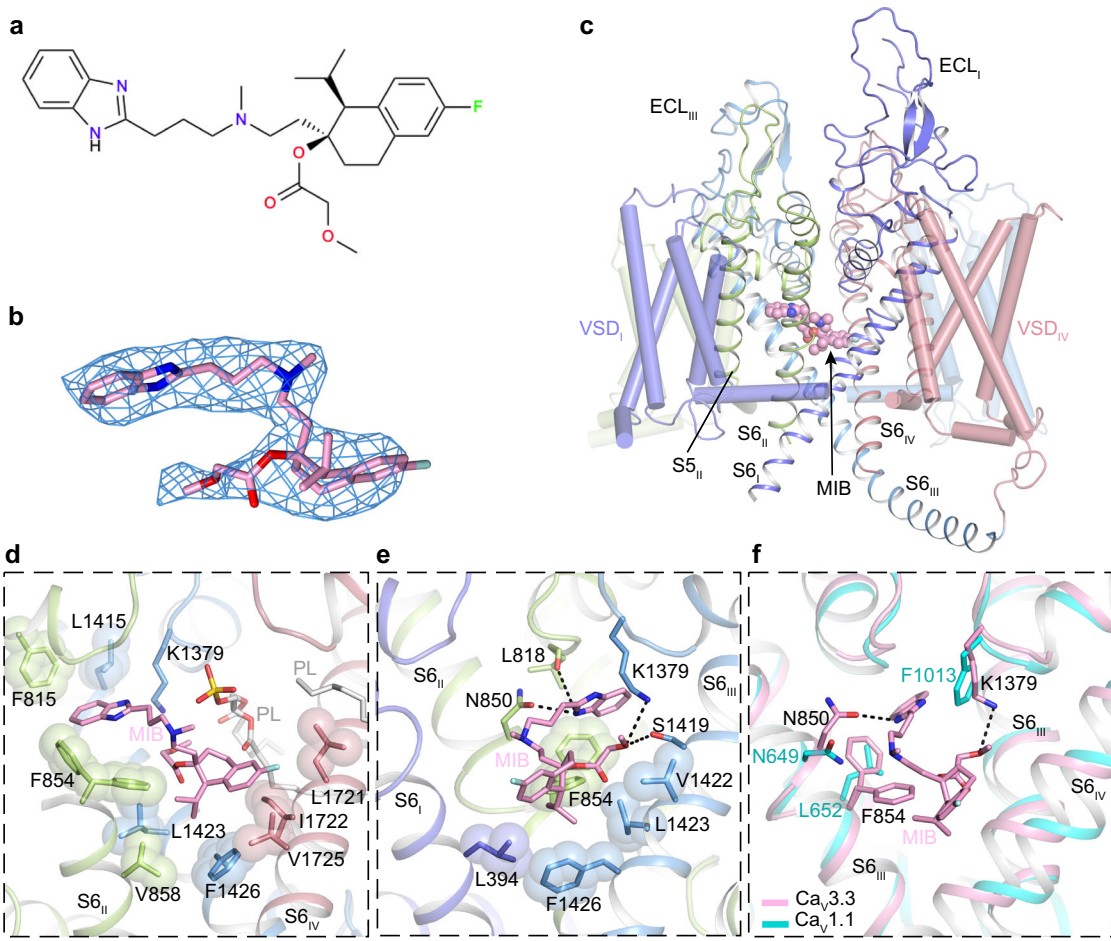

**Fig. 3 Structure basis for blockade of Ca$_V$3.3 by mibefradil. a** Chemical structures of MIB. **b** The cryo-EM density shown in blue mesh for MIB in sticks. **c** The overall structure of the Ca$_V$3.3$^{MIB}$ complex. The domains of Ca$_V$3.3$^{MIB}$ are colored as D$_I$ in purple, D$_{II}$ in green, D$_{III}$ in blue, and D$_{IV}$ in salmon. The MIB in the pore domain is presented as pink spheres. **d**, **e** Detailed binding sites for MIB showing interactions between MIB and Ca$_V$3.3. The side chains of key residues are displayed on sticks and the hydrophobic side chains are overlaid with transparent surfaces. Black dashed lines indicated potential hydrogen bonds. Phospholipids entering through fenestrations are shown in gray. **f** Comparison of the MIB binding sites of Ca$_V$3.3 (pink) with Ca$_V$1.1 (PDB ID: 5GJV) (neon green).

and quaternary ammonium groups (Fig. 4a). We determined the Ca$_V$3.3$^{OB}$ complex at 3.6-Å resolution (Supplementary Fig. 3 and Supplementary Table 1). A single OB molecule binds in the central cavity and close to the selectivity filter. The N-octyloxy and benzoyl groups of OB penetrate through the D$_{II}$–D$_{III}$ fenestration site, leaving the N-octyloxy group outside of the cavity and suggesting that OB may access its binding site by entering through the D$_{II}$–D$_{III}$ fenestration of the Ca$_V$3.3 complex (Fig. 4c–e). The two groups inside the cavity closely pack with surrounding hydrophobic residues from P1, S6$_{II}$, and S6$_{III}$ helices (Fig. 4c-d). The positively charged diethyl methylamine group extends toward the intracellular gate and thus stabilizes the channel in the inactivation state. The phospholipid from the D$_{II}$–D$_{III}$ fenestration site also contributes to the stability of the OB molecule (Fig. 4d). A similar binding pocket was identified in the Ca$_V$1.1 structure, and appears to be compatible with the OB binding (Fig. 4e). However, as mentioned early, the F854 and K1379 residues in Ca$_V$3.3 are substituted by L652 and F1013 in Ca$_V$1.1, respectively, consistent with the selective inhibition of Ca$_V$3.3 over L-type Ca$_V$ channels by OB (Fig. 4e).

**Mechanism of pimozide antagonism.** The diphenylbutylpiperidines pimozide (PMZ) is a clinically approved antipsychotic drug by acting as an antagonist of dopamine receptors[48,49]. It has also

been shown to potently block T-type current in various cell types, with blocking potency parallel its potency as the dopamine receptor antagonist, suggesting that blockade of T-type calcium channels probably is important for its therapeutic efficacy[49]. A PMZ molecule is composed of benzimidazole, piperidinyl, and two fluorophenyl groups (Fig. 5a). Our Ca$_V$3.3$^{PMZ}$ complex structure is determined at 3.6-Å resolution and elucidates that a single PMZ molecule is located in the central cavity formed by S6 helices (Fig. 5b, c and Supplementary Fig. 3). Similar to MIB, the benzimidazole group partially penetrate into the D$_{II}$–D$_{III}$ fenestration site and stabilized by hydrophobic interactions with residues F815$^{P1/II}$, F854$^{S6/II}$, and L1415$^{S6/III}$. Residues N850$^{S6/II}$ and L818$^{P1/II}$ also stabilize the benzimidazole group by forming hydrogen bonds with its amide group (Fig. 5d). The diphenylbutylpiperidine moiety is posed close to the intracellular gate, and its binding relies on extensive hydrophobic interactions with adjacent residues, such as V858, L1423, F1426, I1722, and V1725. One of the fluorophenyl groups is positioned proximal to the S6$_{III}$ helix and point toward the space between residues L1423 and F1426. Consequently, F1426 slightly moves toward the intracellular side, and the intracellular part of the S6$_{III}$ helix bends a little compared with that in the Ca$_V$3.3$^{apo}$ structure (Fig. 5e).

In addition to the diphenylbutylpiperidines pimozide, other antagonists of D2 dopamine receptor, including diphenyldiperazine flunarizine and butyrophenone haloperidol, are

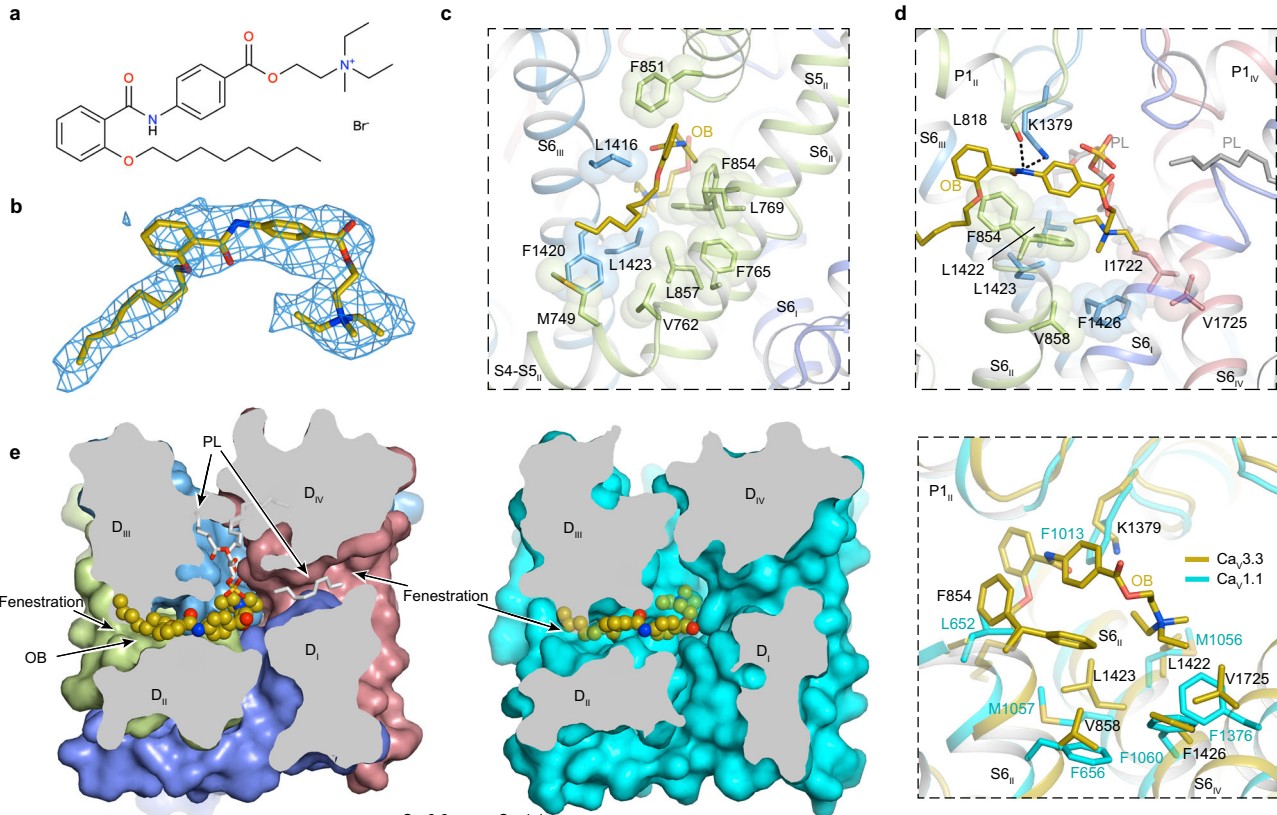

**Fig. 4 Structure basis for blockade of Ca_V3.3 by otilonium bromids. a** Chemical structures of OB. **b** The cryo-EM density shown in blue mesh for OB in sticks. **c** Hydrophobic residues which surrounding the D_II-D_III fenestration site and penetrated by OB are shown in sticks and overlaid with transparent surfaces viewed in facing the D_II-D_III fenestration site. OB is shown as brown sticks. **d** Detailed binding sites of OB in the pore domain. The side chains of key residues are displayed in sticks and overlaid with transparent surfaces. Black dashed lines indicated potential hydrogen bonds. Phospholipids entering through other fenestrations are shown as gray sticks. **e** Binding site of OB in Ca_V3.3^OB with Ca_V1.1 (PDB ID: 5GJV) (neon green) structures. The domains of Ca_V3.3^OB are colored as D_I in purple, D_II in green, D_III in blue, and D_IV in salmon. Extracellular view sectioned below the selectivity filter indicates OB penetrates through the D_II-D_III fenestration site of Ca_V3.3^OB (left panel) and Ca_V1.1(middle panel). Comparison of key residues in the pore domain between Ca_V3.3^OB and Ca_V1.1 is shown in the right panel.

approximately 10-fold less potent at blocking the Ca_V3.3 channels[50]. Compared with pimozide, the haloperidol contains only one fluorophenyl group. Moreover, the benzimidazole group of pimozide is replaced by the chlorophenol group and the phenyl group in haloperidol and flunarizine, respectively[21]. These structural substitutions would abolish some favorable interactions critical for pimozide binding and thus may lead to their distinct ability to block T-type calcium channels. It further supports the notion that the benzimidazole group and fluorophenyl group are essential for the high-potency antagonist activity of pimozide.

Unexpectedly, these structurally different inhibitors preferentially bind at a similar site within the central cavity of the Ca_V3.3 channel, yet interactions critical for their binding are significantly different in detail. These differences provide a framework for the future structure-guided drug design. In particular, the methoxyacetate group of MIB is coordinated by K1379 and S1419. The N-octyloxy and benzoyl groups of OB penetrate through the D_II–D_III fenestration site, and the N-octyloxy group forms extensive interactions with residues located outside of the cavity. The fluorophenyl group of PMZ is posed close to the S6_III helical groove, resulting in a slightly bended S6_III helix. The dichlorobenzene group is mainly stabilized by interaction with L769. Moreover, phospholipids are determined in the central cavity of all structures, entering from the D_III–D_IV and D_I–D_IV fenestrate sites. However, their positions are distinct among these structures, especially the phospholipids are shifted towards the central axis of

the channel in various extent upon inhibitor binding (Supplementary Fig. 6). These phospholipids are likely to be important for inhibitors binding. In the Ca_V3.3^apo, Ca_V3.3^MIB, Ca_V3.3^OB, and Ca_V3.3^PMZ complexes, the F854 side chain exhibits two alternative conformations (F854^A and F854^B). The side-chain of F854^A forms a π−π interaction in a T-shaped configuration with the benzmidazole group of MIB or PMZ, or the benzoyl group of OB (Supplementary Fig. 7a). We speculate that F854 is remarkably important for these drug binding. We mutated F854 to alanine (F854A) and carried out electrophysiological experiment. This mutation does not obviously affect gating properties of the channel, in terms of activation and inactivation, but it does significantly reduce efficacy of these inhibitors at blocking the channels, supporting our structural observations (Supplementary Figs. 7b, c).

## Methods

**Ca_V3.3 expression and purification**. The human Ca_V3.3 cDNA sequence (Uni-Prot ID: Q9P0X4) was cloned into pEG BacMam vector. For Ca_V3.3^EM, residues 1–42 and 2065–2223 were deleted to optimize the protein expression level and stability. An incidental point mutation, E523Q was introduced during the cloning. Electrophysiology experiment indicates that channel properties of the Ca_V3.3^EM and the full-length protein were not changed. Specifically, Ca_V3.3^EM protein was fused with a PreScission protease cleavage site (SNSLEVLFQ/GP) and a C-terminal GFP-Twin strep tag (Supplementary Table 2). HEK293F cells at cell density of ~2 × 10^6 culture were infected with 10% (v/v) of the Ca_V3.3^EM baculoviruses to initiate the transduction. Cells were harvested ~72 h post infection and stored at −80 °C. Cell pellets were resuspended in ice-cold buffer containing 20 mM HEPES (pH 7.5), 150 mM NaCl, 5 mM β-mercaptoethanol, and protease inhibitors

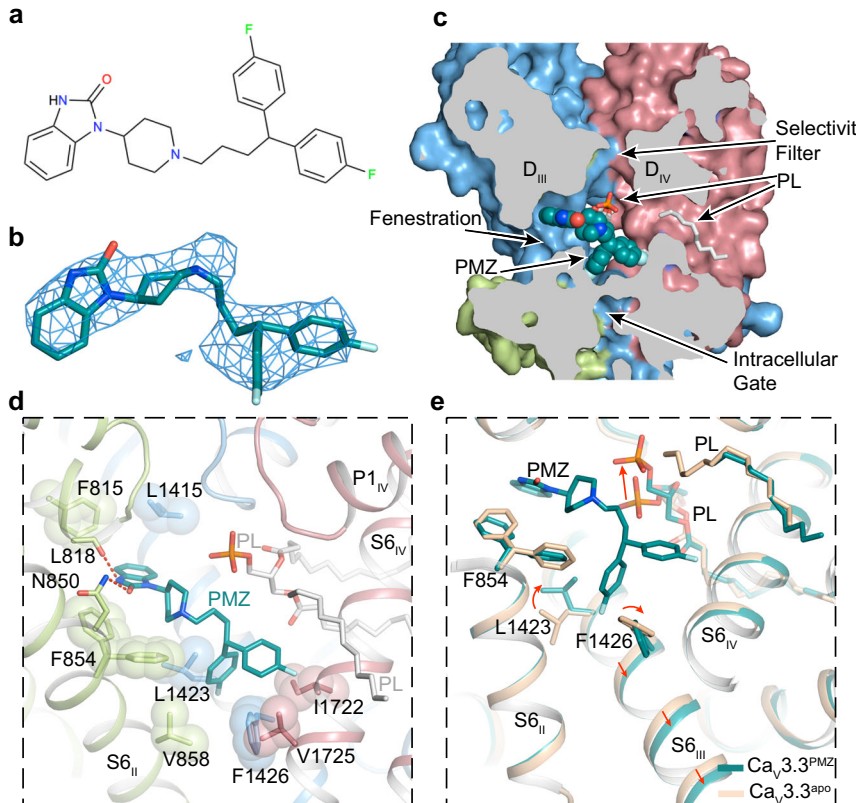

**Fig. 5 Structure basis for blockade of Ca$_V$3.3 by pimozide. a** Chemical structures of pimozide. **b** The cryo-EM density shown in blue mesh for PMZ in sticks. **c** Cross-sectional view, showing the open selectivity filter, fenestrations, pore domain where PMZ (blue spheres) located and intracellular gate. Phospholipids entering through other fenestrations are displayed as white sticks. **d** Detailed binding sites for pimozide. The side chains of key residues are displayed in sticks and the hydrophobic side chains are overlaid with transparent surfaces. Black dashed lines indicated potential hydrogen bonds. **e** Comparison of Ca$_V$3.3$^{PMZ}$ with Ca$_V$3.3$^{apo}$ (wheat). The shifts of the backbone of S6$_{III}$ helix and side chain of L1423 and F1426 and phosphate group of the phospholipid are indicated by red arrows.

Cocktail (Roche, Swiss). The membrane was collected by ultracentrifugation at 4 °C (100,000 *g* for 1 h). Ca$_V$3.3$^{EM}$ protein was extracted with buffer containing 20 mM HEPES (pH 7.5), 150 mM NaCl, 5 mM MgCl$_2$, 1 mM ATP, 5 mM β-mercaptoethanol, 1% (w/v) n-dodecyl-β-D-maltopyranoside (DDM, Anatrace), and 0.2% (w/v) cholesteryl hemisuccinate Tris salt (CHS, Anatrace) for 2 h at 4 °C. Insoluble material was removed by centrifugation (100,000 *g*, 1 h). The supernatant was filtered and passed through Streptactin Beads at 4 °C. The beads were washed with 20 mM HEPES (pH 7.5), 150 mM NaCl, 5 mM β-mercaptoethanol, 5 mM MgCl$_2$, 1 mM ATP and 0.05% (w/v) glyco-diosgenin (GDN) (Anatrace). Then, Ca$_V$3.3$^{EM}$ protein was eluted with buffer containing 5 mM desthiobiotin. Ca$_V$3.3$^{EM}$ protein was further purified by gel filtration chromatography (Superose 6, 10/300) with a buffer containing 20 mM HEPES (pH 7.5), 150 mM NaCl, 5 mM β-mercaptoethanol, and 0.01% (w/v) GDN. Peak fractions were pooled and concentrated to ~4 mg/ml for cryo-EM grids preparation.

**Cryo-EM sample preparation and data acquisition.** Quantifoil R1.2/1.3 Cu 300 mesh grids were glow discharged for 60 s in H$_2$-O$_2$ condition. 4 μL of Ca$_V$3.3$^{EM}$ at ~4 mg/ml was applied to the grid followed by blotting for 5.0 s at 100% humidity and 4 °C, and flash-frozen in liquid ethane using a Vitrobot Mark IV (Thermo Fisher Scientific, USA).

Grids were imaged with a 300 kV Titan Krios (Thermo Fisher Scientific, USA) equipped with a K2 Summit direct electron detector (Gatan, USA) and a GIF-Quantum energy filter. The slit width was set to 20 eV. A calibrated magnification of 105,000× was used, yielded a pixel size of 1.36 Å on images. The defocus range was set to between −1.2 and −2.2 μm. All movie stacks were collected using SerialEM[51] under a dose rate of 9.1–9.4 e$^-$/pixel/s with a total exposure time of 11.4 s, and dose-fractioned to 32 frames, resulting in a total dose of 60 e$^-$/Å2.

**Single-particle cryo-EM data processing.** As for the data processing of Ca$_V$3.3, a total of 1841 movie stacks were collected and motion-corrected[52]. Parameters of the contrast transfer function (CTF) were estimated using Gctf[53]. A total of 1,140,797 particles were automatically picked using Gautomatch and Template Picker (cryoSPARC)[54]. Initial references were calculated using Ab-initio

Reconstruction in cryoSPARC[54]. All further steps of image processing were performed in RELION 3.1[55]. Two rounds of guided multi-reference 3D classification were performed against one good and seven biased references. The first class (55.0%) clearly showed transmembrane helices and was thus selected for further processing. The second round of guided 3D classification also generated a good class (68.4%) which displayed clearly resolved transmembrane helices, and was submitted for following 3D refinement, yielding a 3.9-Å map. Another local-search 3D classification (angular search range = 15°) was applied to improve the quality of the map. The class 1 (21.9%) among the resulting 8 classes solely exhibited high-resolution structural features and was subjected to subsequent Bayesian Polish, CTF refinement, and 3D auto refinement. The final map was reported at 3.3 Å according to the golden-standard *Fourier* shell correlation (GSFSC) criterion.

A similar strategy was applied in the data processing of Ca$_V$3.3$^{PMZ}$, Ca$_V$3.3$^{MIB}$, and Ca$_V$3.3$^{OB}$ except for the last round of 3D classification. Specifically, a total of 1,623,565, 875,466 and 1,684,039 particles were picked form 3909, 2220 and 2766 micrographs respectively. A following round of focused 3D classification was performed without image alignment, while applying a mask excluding micelles. Classes displaying high-resolution structural features and recognizable density of small molecules were submitted to following procedures, yielding 3D reconstructions at 3.6 Å, 3.9 Å, and 3.6 Å resolution respectively, according to the GSFSC criterion.

**Model building.** To build the atomic model of Ca$_V$3.3$^{apo}$, the structure of Ca$_V$3.1(PDB ID: 6KZO) was used as an initial template and fitted to the EM map of Ca$_V$3.3$^{apo}$ as a rigid body using the UCSF Chimera[56]. The residues were mutated according to the sequence alignment between Ca$_V$3.3 and Ca$_V$3.1. The model was then manually adjusted in COOT[57] iteratively, including the refinement of main chain and side chains of residues, and addition of residues where the corresponding density showed density features. Phospholipids and cholesterol hemisuccinate molecules were manually placed in the strip-shaped densities in both leaflet of the lipid bilayers and the fenestration of the pore domain of Ca$_V$3.3 channel. Real-space refinement was then performed in the presence of secondary structure and Ramachandran restraints using PHENIX.real_space_refine[58].

For the model building of $Ca_V3.3^{MIB}$, $Ca_V3.3^{OB}$, and $Ca_V3.3^{PMZ}$, the $Ca_V3.3^{apo}$ structure was used as a starting model. The atomic model of $Ca_V3.3$ was fit into the EM maps as a rigid body. The side chains were manually adjusted in COOT[57]. The structure data files (SDFs) of the drugs were manually drawn in ChemDraw, followed by the generation of 3D models and refinement restraints in phenix.ligand_eLBOW. The drug molecules were docked in the EM map and refined according to the corresponding density. All the manually adjusted models were then subjected to the real-space refinement using PHENIX.real_space_refine.

All of the figures were prepared with Pymol[59], UCSF Chimera[56].

**Whole-cell voltage-clamp recordings**. HEK 293T cells were cultured with Dulbecco's Modified Eagle Medium (Gibco) added 15% (v/v) fetal bovine serum (FBS) (PAN-Biotech) at 37 °C with 5% $CO_2$. The cells were grown in the culture dishes ($d = 3.5$ cm) (Thermo Fisher Scientific) for 24 h and then transiently transfected with 1 μg control or mutant plasmids expressing GFP-fused human $Ca_V3.3$ using 0.7 μg Lipofectamine 2000 Reagent (Thermo Fisher Scientific). Experiments were performed 12–24 h post transfection at room temperature (21–25 °C). In brief, cells were placed on a glass chamber containing 105 mM NaCl, 10 mM $BaCl_2$, 10 mM HEPES, 10 mM D-Glucose, 30 mM TEA-Cl, 1 mM $MgCl_2$, 5 mM CsCl, (pH = 7.3 with NaOH and osmolarity of ~310 mos $mol^{-1}$). Whole-cell voltage-clamp recordings were made from isolated, GFP-positive cells using 2–5 MΩ fire polished pipettes (Sutter Instrument) when filled with standard internal solution, containing 135 mM K-Gluconate, 10 mM HEPES, 5 mM EGTA, 2 mM $MgCl_2$, 5 mM NaCl, 4 mM Mg-ATP, (pH = 7.2 with CsOH and osmolarity of ~295 mos $mol^{-1}$). Whole-cell currents were recorded using an EPC-10 amplifier (HEKA Electronic) at 20 kHz sample rate and was low pass filtered at 5 kHz. The series resistance was 2–7 MΩ and was compensated 80–90%. The data was acquired by PatchMaster program (HEKA Electronic).

To characterize the activation properties of $Ca_V3.3$ channels, cells were held at −100 mV and then a series of 500 ms voltage steps from −100 mV to +30 mV in 10 mV increments were applied. The inactivation properties of $Ca_V3.3$ channels were assessed with a 3.6 s holding-voltages ranging from −110 mV to 0 mV (10 mV increments) followed by a 200 ms test pulse at −20 mV.

All data reported as mean ± SEM. No statistical methods were used to predetermine sample sizes but our sample sizes are similar to those reported previously in the field[29,31,60]. Data analyses were performed using Origin 2019b (Origin Lab Corporation), Excel 2016 (Microsoft), GraphPad Prism 6 (GraphPad Software, Inc.), and Adobe illustrator 2018 (Adobe Systems Incorporated). Steady-state activation curves were generated using a Boltzmann equation.

$$\frac{g}{g_{max}} = \frac{1}{1 + \exp(V - V_{0.5})/k} \quad (1)$$

where $g$ is the conductance, $g_{max}$ is the maximal conductance of $Ca_V3.3$ during test pulse, $V$ is the test potential, $V_{0.5}$ is the half-maximal activation potential and $k$ is the slope factor.

Steady-state inactivation curves were generated using a Boltzmann equation.

$$\frac{I}{I_{max}} = \frac{1}{1 + \exp(V - V_{0.5})/k} \quad (2)$$

where $I$ is the current at indicated test pulse, Imax is the maximal current of $Ca_V3.3$ during test-pulse, $V$ is the test potential, $V_{0.5}$ is the half-maximal inactivation potential and $k$ is the slope factor.

Inhibition curves were generated using a Hill equation.

$$\frac{I}{I_{max}} = \frac{1}{1 + 10^{(\log IC50 - [C]) \times H}} \quad (3)$$

where $I$ is the current at different drug concentrations, $I_{max}$ is the maximal current of $Ca_V3.3$ without drug is applied, $[C]$ is the logarithmic concentration of drugs, $IC_{50}$ is the half-maximal inhibitory concentration and $H$ is the Hill coefficient. Statistical significance ($P < 0.05$) was determined using unpaired Student's $t$ tests or one-way ANOVA with Tukey's post hoc test.

**Reporting summary**. Further information on research design is available in the Nature Research Reporting Summary linked to this article.

## Data availability

The three-dimensional cryo-EM density maps of $Ca_V3.3^{apo}$, $Ca_V3.3^{MIB}$, $Ca_V3.3^{OB}$ and $Ca_V3.3^{PMZ}$ have been deposited in the EM Database under the accession codes EMD-32584, EMD-32585, EMD-32586, and EMD-32587, respectively. The corresponding coordinates for these complexes have been deposited in Protein Data Bank under accession codes 7WLI [https://doi.org/10.2210/pdb7WLI/pdb], 7WLJ [https://doi.org/10.2210/pdb7WLJ/pdb], 7WLK [https://doi.org/10.2210/pdb7WLK/pdb] and 7WLL [https://doi.org/10.2210/pdb7WLL/pdb], respectively. Source data are provided with this paper.

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

## Acknowledgements

We thank X. Huang, B. Zhu, and other staff members at the Center for Biological Imaging (CBI), Core Facilities for Protein Science at the Institute of Biophysics, Chinese Academy of Science (IBP, CAS) for the support in cryo-EM data collection. We thank Yan Wu for his research assistant service. This work is funded by National Key Research and Development Program of China (grant 2021YFA1301501 to Y.Z.), Chinese Academy of Sciences Strategic Priority Research Program (grant XDB37030304 to Y.Z.), the National Natural Science Foundation of China (grant 92157102 to Y.Z.), Chinese National Programs for Brain Science and Brain-like Intelligence Technology (grant 2021ZD0202102 to Z.H.), the National Natural Science Foundation of China (grant 81371432 to Z.H.), and the Youth Innovation Promotion Association of the Chinese Academy of Sciences (grant 2022089 to L.H.).

## Author contributions

Y.Z. conceived the project and supervised the research. L.H. carried out molecular cloning and cell biology experiments. L.H. expressed and purified protein samples. Y.D. and L.H. prepared sample for cryo-EM study. Z.Y., L.H., and Y.G carried out cryo-EM data collection. Z.Y. and Y.Z. processed the cryo-EM data. Z.Y., Y.Z., and Q.C. prepared figures. Z.Y. and Y.W. built and refined the atomic model. Y.Z., Z.Y., and B.H. analyzed the structures and designed mutants. Y.Z., Z.H., X.W., Z.G., C.Z., L.S., and X.M. designed and performed electrophysiological experiments. Y.Z., L.H., and Y.D. prepared the manuscript with input from all authors.

## Competing interests

The authors declare no competing interests.
