## [Peer Review File · Nature Communications]

Structure, gating and pharmacology of human CaV3.3 channelREVIEWER COMMENTS

Reviewer #1 (Remarks to the Author):

This manuscript reports a structure of the CaV3.3 calcium channel determined using cryo-electron microscopy (cryo-EM) and molecular modelling. Channel was truncated for cryo-EM studies without affecting voltage dependencies of channel activation and inactivation. In addition to model of the truncated channel itself, structure with three different T-type channel blockers mibefradil (MIB), otilonium bromid (OB) and pimozide (PMZ) was resolved. Authors discuss implications of their model for channel gating and for drug binding into the channel pore. Manuscript adds new interesting facts to current knowledge about gating mechanism of low-voltage activated calcium channels.

Experimental procedures are described in sufficient detail and are proper for the aim of experiments. No statistical method was used for analysis of the data from electrophysiological measurements, however, it was also not necessary for the purpose of this investigation.

Most interesting is the suggested interpretation of low voltage threshold for CaV3 channels activation. Authors suggest that positively charged cytoplasmic part of IIS6 helix may act as an additional voltage sensor. While reported features of CaV3.3 gating and drug binding are interesting and novel, their presentation is partly confusing and need an improvement.

Specific questions:

1. You suggested that cytoplasmic part of IIS6 helix may act as an additional voltage sensor and that it may be partly responsible for low voltage threshold for activation. This hypothesis is in line with a suggestion that for opening of the conducting pore of the CaV channels activation of S4 segments (voltage sensors) in all four domains is not necessary. Please address this possibility. Any relation to putative gating brake in I-II loop? Please discuss.

2. When you mutated cytoplasmic region of the IIS6 segment, voltage dependence of channel activation was shifted to the more positive voltages. Was also current activation kinetics altered?

3. Please provide rationale for comparison of the CaV3.3 channel with the CaV2.2 channel. Is it necessary part of the manuscript? Why did you look for an interaction of the CaV3.3 with alpha2delta subunit? You should either explain the rationale or leave out this part, which is confusing.

4. Mibefradil is not very specific T-type channel blocker, IC50 for L-type calcium channel (CaV1.2) is just 10-fold higher (Mehrke et al. 1994; Bezprozvanny and Tsien 1995). Please correct your statement. Further, for the part of manuscript focused on drug binding to the channel pore you compared CaV3.3 structure with CaV1.1 structure. Why this channel was chosen? It would be more appropriate to choose CaV1.2 for comparison, as this channel is comparably blocked by mibefradil, otilonium bromide, and pimozide. CaV1.1 channel has kinetics and voltage dependence of activation very different from all other voltage dependent calcium channels. If this was the reason to choose it for comparison, it should focus on channel gating, not on drug binding.

5. In data availability section accession codes are missing.

Reviewer #2 (Remarks to the Author):

This is a nice manuscript reporting the first structures of human Cav3.3 channel that belong to the T-type calcium channels, which are expressed throughout the nervous, endocrine and cardiovascular systems and implicated in numerous pathologies, including neurological disorders, hypertension and cardiac arrhythmia. This is not surprising, therefore, that Cav3.3 channels represent an important drug target. He et al. combines different approaches, including single-particle cryo-EM, mutagenesis and functional recordings, to solve human Cav3.3 channel structures alone and in complex with several ion channel blockers. The authors clearly identify binding site of these blockers and propose the mechanism of Cav3.3 channel block. They also reveal molecular details of blocker-channel interaction that will help to develop better future drugs. The study is of high quality, the results are presented clearly and I only have minor suggestions to further improve this manuscript.

Minor suggestions

1. Line 73. Shouldn't that be "CaV3.1" instead of "CaV3.3"?
2. Lines 87-89. The fact that the introduced modifications do not alter voltage-dependencies of activation and steady-state inactivation does not prove that all properties of the channel remain unaltered. For example, in Supplementary Figure 1, currents for Cav3.3EM appear smaller than currents for wild type Cav3.3. Did authors measure channel open probability or single-channel conductance? Unless the authors want to get into deeper characterization of Cav3.3 function, I would suggest to say that the modifications did not alter voltage-dependencies of activation and steady-state inactivation and not to generalize conclusions to all channel properties (like it is done on lines 266-267).
3. Line 96. The symbol of Angstroms is missing.

4. Lines 110-113. The statement about negatively charged residues determining selectivity for Ca²⁺ ions requires citations.
5. Line 187. Please label L1721 in Figure 3d-e.
6. Line 188. Typo: "alone" should be replaced with "along".
7. Figure 2d legend. Please provide the PDB accession number for the utilized Cav2.2 coordinates.
8. Figure 3e. L818 looks like a serine in this panel. Is it missing the side chain?
9. Figure 3f and 4e legends. Please provide the PDB accession number for the utilized Cav1.1 coordinates.
10. Supplementary Figure 5 legend. Please provide the PDB accession numbers for the utilized Cav3.1 and Cav2.2 coordinates.

Reviewer #3 (Remarks to the Author):

The authors address a key question in both structural biology as well as they provide an advance in understanding the structure/function of T-type channels (Cav3.3). While the manuscript is well written, there are still some comments/clarifications that need to be addressed before publication.

First, at the end of the introduction it seems there is a confusion between Cav3.3(this study) and Cav3.1. Secondly, the authors discuss mutations of the S6 and they discuss rigidity/flexibility. However, they did not provide structure models/simulations supporting this hypothesis. Additionally, the authors mention a comparison with Cav2.2, however, there seems to be no quantitative comparison such as RMSD, etc.. In general a quantitative overall comparison of all calcium channel structures would be valuable (Cav1.1, Cav2.2 and Cav3.1).

I could also not find the PDB structure. If possible could the authors provide the PDB structure, as well as the electron density map?

Figure 1 would profit from more labels, as the figure caption is rather confusing. There is an inconsistency about the selectivity filter, the selectivity filter is composed of four negatively charged residues "The selectivity filter ring of four positively charged residues from the four domains are shown in sticks".

Reviewer #1 (Remarks to the Author):

This manuscript reports a structure of the Ca_v3.3 calcium channel determined using cryo-electron microscopy (cryo-EM) and molecular modelling. Channel was truncated for cryo-EM studies without affecting voltage dependencies of channel activation and inactivation. In addition to model of the truncated channel itself, structure with three different T-type channel blockers mibefradil (MIB), otilonium bromid (OB) and pimozide (PMZ) was resolved. Authors discuss implications of their model for channel gating and for drug binding into the channel pore. Manuscript adds new interesting facts to current knowledge about gating mechanism of low-voltage activated calcium channels.

Experimental procedures are described in sufficient detail and are proper for the aim of experiments. No statistical method was used for analysis of the data from electrophysiological measurements, however, it was also not necessary for the purpose of this investigation.

Most interesting is the suggested interpretation of low voltage threshold for Ca_v3 channels activation. Authors suggest that positively charged cytoplasmic part of IIS6 helix may act as an additional voltage sensor. While reported features of Ca_v3.3 gating and drug binding are interesting and novel, their presentation is partly confusing and need an improvement.

Reply: We appreciate very much the reviewer's positive comment and his/her suggestions for improving our manuscript.

Specific questions:

1. You suggested that cytoplasmic part of IIS6 helix may act as an additional voltage sensor and that it may be partly responsible for low voltage threshold for activation. This hypothesis is in line with a suggestion that for opening of the conducting pore of the Ca_v channels activation of S4 segments (voltage sensors) in all four domains is not necessary. Please address this possibility. Any relation to putative gating brake in I-II loop? Please discuss.

Reply: We appreciate the reviewer's comment. We agree that the opening of the ion-conducting pore of Ca_v channel does not require activation of S4 segments in all four domains, which have been proved in different studies. For example, study on high-voltage activated Ca_v1.2 shows that VSD_{II} and VSD_{III} are necessary to open the pore, while the VSD_{IV} does not appear to participate in channel activation [1]. As for the low-voltage activated (LVA) Ca_v3.1 channel, the VSD_I play the most important role for channel opening while VSD_{II} and VSD_{III} play lesser roles [2]. The gating brake consists of 60 residues, which was first determined in Ca_v3.2 channel [3]. It is located within the I-II loop close to the S6_I helix, and is conserved in all LVA type channels [3]. Previous studies indicate the gating brake potentially forms contacts with the gating machinery, thereby stabilizing a resting state of T-channels [3]. When membrane potential is depolarized, it dissociates from the channel, thus allowing the S6 segments to spread open and Ca²⁺ ions to flow through [3]. In the absence of the gating brake, the voltage dependence of activation of Ca_v3.3 is shifted towards more negative membrane potentials [4]. Taken together, we believe the VSDs, gating break and the positively charged extended S6_{III} are all important for activation of the T-type Ca_v channel. We speculate the extended S6_{III} helix in synergy with the VSD and gating break to bias the channel towards opening at negative membrane potential. However, the gating brake cannot be visualized in our structure, further structural and functional studies are required to fully understand activation mechanism of the T-type channels.

We also added a short discussion in our revised manuscript in the line 172-178, it reads "Previous reports have shown that the opening of the ion-conductive pore of T-type calcium channels does not require activation of all four VSDs and the gating brake located within the I-II loop plays an important role in regulating the channel opening³⁸⁻⁴¹. We speculate this extended and positively charged S6_{III} helix may cooperate with VSDs and gating brake to modulate the gating mechanism of T-type channel. However, more studies are required to fully understand this potential synergistic regulation mechanism."

[1] Pantazis, A., Savalli, N., Sigg, D., Neely, A. & Olcese, R. Functional heterogeneity of the four voltage sensors of a human L-type calcium channel. *Proc Natl Acad Sci U S A* 111, 18381-18386, doi:10.1073/pnas.1411127112 (2014).

[2] Jurkovicova-Tarabova, B., Mackova, K., Moravcikova, L., Karmazinova, M. & Lacinova, L. Role of individual S4 segments in

gating of Ca(v)3.1 T-type calcium channel by voltage. *Channels (Austin)* 12, 378-387, doi:10.1080/19336950.2018.1543520 (2018).

[3] Perez-Reyes, E. Characterization of the gating brake in the I-II loop of CaV3 T-type calcium channels. *Channels (Austin)* 4, 453-458, doi:10.4161/chan.4.6.12889 (2010).

[4] Karmažínová, M., Baumgart, J. P., Perez-Reyes, E. & Lacinová, L. The voltage dependence of gating currents of the neuronal Ca(v)3.3 channel is determined by the gating brake in the I-II loop. *Pflugers Arch* 461, 461-468, doi:10.1007/s00424-011-0937-2 (2011).

2. When you mutated cytoplasmic region of the IIS6 segment, voltage dependence of channel activation was shifted to the more positive voltages. Was also current activation kinetics altered?

Reply: We thank the reviewer for raising this point. To investigate the current activation kinetics of mutated Ca_v3.3, we have analyzed the activation time course (τ) for Ca_v3.3^{WT}, Ca_v3.3^{12Q} and Ca_v3.3^{3G} by fitting 20-80% of the peak currents with a single exponential function ($y = y_0 + Ae^{-x/\tau}$) at the voltage of maximal activation (V_{max}) (Figure 1*). The activation time constant of Ca_v3.3^{WT} was ~ 11ms, which was consistent with previous study [5]. In addition, the time constant for Ca_v3.3^{12Q} and Ca_v3.3^{3G} was nearly identical to Ca_v3.3^{WT}.

[5] El Ghaleb, Y., Schneeberger, P. E., Fernández-Quintero, M. L., Geisler, S. M., Pelizzari, S., Polstra, A. M., van Hagen, J. M., Denecke, J., Campiglio, M., Liedl, K. R., Stevens, C. A., Person, R. E., Rentas, S., Marsh, E. D., Conlin, L. K., Tuluc, P., Kutsche, K., & Flucher, B. E. CACNA11 gain-of-function mutations differentially affect channel gating and cause neurodevelopmental disorders. *Brain* 144(7), 2092–2106, doi:10.1093/brain/awab101 (2021).

Figure 1*. The activation time constants of the Ca_v3.3^{WT} (black), Ca_v3.3^{12Q} (blue) and Ca_v3.3^{3G}(red). They are generated by fitting 20-80% of the peak currents with a single exponential function $y = y_0 + Ae^{-x/\tau}$.

3. Please provide rationale for comparison of the Ca_v3.3 channel with the Ca_v2.2 channel. Is it necessary part of the manuscript? Why did you look for an interaction of the Ca_v3.3 with alpha2delta subunit? You should either explain the rationale or leave out this part, which is confusing.

Reply: We appreciate the reviewer's comment. The structure of the Ca_v3.3 channel contains an extended S6 helix from domain III (S6_{III}), with a positive charged region protruding into the cytosol (S6^{Cyto}). According to the sequence alignment, the S6^{Cyto} is exclusively conserved in the T type Ca_v channels. To understand its functional roles, we also compared structures of the Ca_v3.3 with the high-voltage activated Ca_v2.2 channel. It turns out that the positive charge S6^{Cyto} is indeed only present in the Ca_v3.3 but not in Ca_v2.2 channel, which inspired us to speculate this unique S6^{Cyto} probably is crucial for T type channel activation at low voltage. Our functional studies also confirmed our speculations.

We didn't try to look for the interactions between Ca_v3.3 with $\alpha 2\delta$ subunit. We just want to interpret why the $\alpha 2\delta$ subunit could not associate with the T-type channels. This is because the extracellular loops of Ca_v3.3 and Ca_v2.2 adopt significantly distinct conformations.

4. Mibefradil is not very specific T-type channel blocker, IC₅₀ for L-type calcium channel (Ca_v1.2) is just 10-fold

higher (Mehrke et al. 1994; Bezprozvanny and Tsien 1995). Please correct your statement.

Reply: We thank the reviewer for pointing this out. We adjusted related sentences and incorporated these two references in the revised manuscript.

42. Bezprozvanny, I. & Tsien, R. W. Voltage-dependent blockade of diverse types of voltage-gated Ca²⁺ channels expressed in *Xenopus* oocytes by the Ca²⁺ channel antagonist mibefradil (Ro 40-5967). *Mol Pharmacol* 48, 540-549 (1995).

43. Mehrke, G., Zong, X. G., Flockerzi, V. & Hofmann, F. The Ca⁽⁺⁺⁾-channel blocker Ro 40-5967 blocks differently T-type and L-type Ca⁺⁺ channels. *J Pharmacol Exp Ther* 271, 1483-1488 (1994).

In the revised line 180, it now reads “Mibefradil (MIB) is a benzimidazolyl-substituted tetraline derivative that act as a higher affinity blocker for T-type Ca_v channels than for HVA L-type Ca_v channels^{20,42,43}.”.

In the revised line 184, it now reads “However, it is still desirable to understand molecular details how mibefradil blocks T-type Ca_v channels with higher affinity.”.

In the revised line 200, it now reads “MIB has also been characterized to block HVA L-type Ca_v channels (e.g. Ca_v1.1 and Ca_v1.2), yet with much lower efficacy (approximately 10- to 15-fold lower)^{20,42,43}.”.

Further, for the part of manuscript focused on drug binding to the channel pore you compared Ca_v3.3 structure with Ca_v1.1 structure. Why this channel was chosen? It would be more appropriate to choose Ca_v1.2 for comparison, as this channel is comparably blocked by mibefradil, otilonium bromide, and pimozone. Ca_v1.1 channel has kinetics and voltage dependence of activation very different from all other voltage dependent calcium channels. If this was the reason to choose it for comparison, it should focus on channel gating, not on drug binding.

Reply: We appreciate the reviewer 's suggestions and agree with the reviewer in that Ca_v1.2 would be more appropriate for comparison. While only three different subtypes of Ca_v structures have been reported currently, including L-type Ca_v1.1, N-type Ca_v2.2 and T-type Ca_v3.1. The structure of L-type Ca_v1.2 channel has not been determined yet. So we have to use L-type Ca_v1.1 channel for structure comparison. Moreover, according to the sequence alignment of in pore domain (Figure 2*), the Ca_v1.1 and Ca_v1.2 channels share high sequence identity in the pore domain. The residues involved in the MIB binding site are conserved. Therefore, we believe that it is convincing to study binding specificity of drug using structural comparison between Ca_v3.3 and Ca_v1.1 channels.

Figure 2*. Comparison of the MIB binding sites of Ca_v3.3 with Ca_v1.2. a. Sequence alignment of Ca_v1.1 and Ca_v1.2 in the pore domain. The conserved residues are shaded in grey. The key residues involved in drug binding are shaded in blue. b. Comparison of the MIB binding sites of Ca_v3.3 (pink) with Ca_v1.1 (PDB ID: 5GJV) (neon green).

5. In data availability section accession codes are missing.

Reply: We thank the reviewer for pointing this out. The accession codes have been updated in the revised manuscript now (Line 425-429). It reads “The three-dimensional cryo-EM density maps of Ca_v3.3^{apo}, Ca_v3.3^{MIB}, Ca_v3.3^{OB} and Ca_v3.3^{PMZ} have been deposited in the EM Database under the accession codes EMD-32584, EMD-32585, EMD-32586, and EMD-32587, respectively. The corresponding coordinates for these complexes have been deposited in Protein Data Bank under accession codes 7WLI, 7WLJ, 7WLK and 7WLL, respectively.”.

Reviewer #2 (Remarks to the Author):

This is a nice manuscript reporting the first structures of human Ca_v3.3 channel that belong to the T-type calcium channels, which are expressed throughout the nervous, endocrine and cardiovascular systems and implicated in numerous pathologies, including neurological disorders, hypertension and cardiac arrhythmia. This is not surprising, therefore, that Ca_v3.3 channels represent an important drug target. He et al. combines different approaches, including single-particle cryo-EM, mutagenesis and functional recordings, to solve human Ca_v3.3 channel structures alone and in complex with several ion channel blockers. The authors clearly identify binding site of these blockers and propose the mechanism of Ca_v3.3 channel block. They also reveal molecular details of blocker-channel interaction that will help to develop better future drugs. The study is of high quality, the results are presented clearly and I only have minor suggestions to further improve this manuscript.

Reply: We appreciate the reviewer's positive comments very much.

Minor suggestions

1. Line 73. Shouldn't that be "Ca_v3.1" instead of "Ca_v3.3"?

Reply: Thank you for pointing this out. We revised the sentence in the revised manuscript. In the revised line 70, it now reads "Moreover, despite the structure of the Ca_v3.1 has been determined, the mechanism for activating T-type Ca_v channels at low voltages is still elusive."

2. Lines 87-89. The fact that the introduced modifications do not alter voltage-dependencies of activation and steady-state inactivation does not prove that all properties of the channel remain unaltered. For example, in Supplementary Figure 1, currents for Ca_v3.3EM appear smaller than currents for wild type Ca_v3.3. Did authors measure channel open probability or single-channel conductance? Unless the authors want to get into deeper characterization of Ca_v3.3 function, I would suggest to say that the modifications did not alter voltage-dependencies of activation and steady-state inactivation and not to generalize conclusions to all channel properties (like it is done on lines 266-267).

Reply: Thanks for your kind reminders. We did not measure channel open probability or single-channel conductance. We have revised related sentence in line 85 and it now reads "Whole-cell patch-clamp experiment confirmed that these modifications on the Ca_v3.3 construct resulted in similar voltage dependency of the activation and steady-state inactivation compared to full-length wild-type Ca_v3.3 (Fig. 1 and Supplementary Fig. 1)."

3. Line 96. The symbol of Angstroms is missing.

Reply: We thank the reviewer for pointing this out and have added Angstrom symbols.

4. Lines 110-113. The statement about negatively charged residues determining selectivity for Ca²⁺ ions requires citations.

Reply: Thanks for your kind reminders. We have cited the following references to support our statement (line 111).
33. Yang, J., Ellinor, P. T., Sather, W. A., Zhang, J. F. & Tsien, R. W. Molecular determinants of Ca²⁺ selectivity and ion permeation in L-type Ca²⁺ channels. *Nature* 366, 158-161, doi:10.1038/366158a0 (1993).

34. Talavera, K. et al. Aspartate residues of the Glu-Glu-Asp-Asp (EEDD) pore locus control selectivity and permeation of the T-type Ca(2+) channel alpha(1G). *J Biol Chem* 276, 45628-45635, doi:10.1074/jbc.M103047200 (2001).

5. Line 187. Please label L1721 in Figure 3d-e.

Reply: We thank the reviewer for pointing this out. We have added label L1721 in the Figure 3d. We did not label the L1721 in Figure 3e as it cannot be visualized. (Figure 3*).

Figure 3*. Old and revised versions of figure 3d. **a.** Old versions of figure 3d. **b.** Revised versions of figure 3d. I1721 has been labelled.

6. Line 188. Typo: “alone” should be replaced with “along”.

Reply: We thank the reviewer for pointing out this typo and we have corrected it in our revised manuscript.

7. Figure 2d legend. Please provide the PDB accession number for the utilized $Ca_v2.2$ coordinates.

Reply: Thank you very much for the reminder. The PDB accession number of the utilized $Ca_v2.2$ coordinates is 7VFS. We have provided the PDB ID in the main text and figure legend.

8. Figure 3e. L818 looks like a serine in this panel. Is it missing the side chain?

Reply: Thank you very much for the reminder. We have shown the side chain of L818 in panel d of figure 3e.

Figure 4*. Old and revised versions of figure 3e. **a.** Old versions of figure 3e. **b.** Revised versions of figure 3e. The side chain of L818 has been shown.

9. Figure 3f and 4e legends. Please provide the PDB accession number for the utilized $Ca_v1.1$ coordinates.

Reply: Thank you very much for the reminder. The PDB accession number of the utilized $Ca_v1.1$ coordinate is 5GJV and has been provided in the main text and figure legend.

10. Supplementary Figure 5 legend. Please provide the PDB accession numbers for the utilized $Ca_v3.1$ and $Ca_v2.2$ coordinates.

Reply: Thank you very much for the reminder. We have included PDB ID of $Ca_v3.1$ and $Ca_v2.2$ in the main text and figure legend.

Reviewer #3 (Remarks to the Author):

The authors address a key question in both structural biology as well as they provide an advance in understanding the structure/function of T-type channels ($Ca_v3.3$). While the manuscript is well written, there are still some comments/clarifications that need to be addressed before publication.

Reply: We appreciate reviewer’s positive comment. We have carefully checked throughout the manuscript and made corrections accordingly.

First, at the end of the introduction it seems there is a confusion between Cav3.3 (this study) and Cav3.1.

Reply: Thank you for pointing this out. The revised sentence reads: “Moreover, despite the structure of the Cav3.1 has been determined, the mechanism for activating T-type Cav channels at low voltages is still elusive.”.

Secondly, the authors discuss mutations of the S6 and they discuss rigidity/flexibility. However, they did not provide structure models/simulations supporting this hypothesis.

Reply: We appreciate this comment. To explore functional roles of the positively charged S6^{Cyto} region, we neutralized S6^{Cyto} by substituting all arginine and lysine to glutamine (Cav3.3^{12Q}) and mutated the “1442EAE¹⁴⁴⁴” to “GGG” (Cav3.3^{3G}) to break the extended S6 helix into two segments. We agree with reviewer that structure models/simulations of these mutants would be helpful to support our hypothesis, but according to some previous studies, we believe that our experiments and related discussions are credible with sufficient basis.

1, Previous secondary structure studies found that, according to the helix propensity scale, glycine and proline have the lowest helix propensity [1]; they are the most preferred helix-stop residues according to the analysis of the alpha-helix-stop signal [2]. The arginine, lysine and glutamine have similar helix propensity [1].

2, We also carried out secondary structure predictions on Cav3.3^{WT} and two mutants Cav3.3^{12Q} and Cav3.3^{3G}. For the Cav3.3^{WT} construct, the residues from P1355 to K1463 prefer to form a long helix (S6_{III}) (Figure 5a*), which is consistent with our observations in the Cav3.3 structures. The S6_{III} helix is not altered in the Cav3.3^{12Q} construct (Figure 5b*). However, the original extended S6_{III} helix is stopped at the triple-glycine mutation site and separated into two shorter helical segments (S6_{III}-1 and S6_{III}-2) once the triple-glycine mutation was introduced (Figure 5c*), which is also in line with previous studies suggesting the glycine is a helix-stop residue [1,2].

Figure 5*. Secondary structure prediction of Cav3.3^{WT} (a) and two mutants Cav3.3^{12Q} (b) and Cav3.3^{3G} (c). Residues involved in Cav3.3^{3G} are marked by red triangle.

3, In previous studies, glycine mutations were commonly used to disrupt the helix. For example, the gating brake of the T-type Cav channels has a helix-loop-helix structure. To investigate its functional roles, several residues from the two helices were substituted by alanine and glycine-proline, respectively. By this way, further experiments confirmed the structural integrity of the helix is critical for normal function of the T-type Cav channels [3].

[1] Pace CN, Scholtz JM. A helix propensity scale based on experimental studies of peptides and proteins. *Biophys J*. 1998;75(1):422-427. doi:10.1016/s0006-3495(98)77529-0

[2] Gunasekaran K, Nagarajaram HA, Ramakrishnan C, Balam P. Stereochemical punctuation marks in protein structures: glycine and proline containing helix stop signals. *J Mol Biol*. 1998;275(5):917-932. doi:10.1006/jmbi.1997.1505

[3] Arias-Olguín II, Vitko I, Fortuna M, et al. Characterization of the gating brake in the I-II loop of Ca(v)3.2 T-type Ca(2+) channels. *J Biol Chem*. 2008;283(13):8136-8144. doi:10.1074/jbc.M708761200.

Additionally, the authors mention a comparison with Cav2.2, however, there seems to be no quantitative comparison such as RMSD, etc.. In general a quantitative overall comparison of all calcium channel structures

would valuable (Cav1.1, Cav2.2 and Cav3.1).

Reply: Thank reviewer for raising this point. The structure of the Cav3.3^{apo} was compared with the structures of Cav1.1 and Cav2.2, giving rise to an RMSD of ~2.5 Å for 920 C α -pairs and ~2.8 Å for 930 C α -pairs, respectively. We have added quantitative comparison results in our revised manuscript.

In the revised line 138, it now reads “The Cav3.3 structure is further compared with the structure of a high-voltage activated Cav2.2 channel (PDB ID: 7VFS), giving rise to an RMSD of ~2.8 Å for 930 C α -pairs.”.

In the revised lines 202, it now reads “Superimposition of Cav3.3^{MIB} complex with HVA L-type Cav1.1 channel (PDB ID: 5GJV) resulted in an RMSD of ~2.5 Å for 920 C α -pairs and no obvious steric clash is observed between the MIB and residues from Cav1.1.”.

I could also not find the PDB structure. If possible could the authors provide the PDB structure, as well as the electron density map?

Reply: We have uploaded our coordinates and corresponding maps to Google drive. Please check them using this link: https://drive.google.com/drive/folders/1dQBDp_hFqz5lvSsLjyVunyOhARMI_LV?usp=sharing

Figure 1 would profit from more labels, as the figure caption is rather confusing. There is an inconsistency about the selectivity filter, the selectivity filter is composed of four negatively charged residues "The selectivity filter ring of four positively charged residues from the four domains are shown in sticks".

Reply: We thank reviewer for pointing out this typo. We have corrected it and added more labels in Figure 1. The new figure legend reads “The selectivity filter ring of four negatively charged residues from the four domains are shown in sticks in the upper, and the intracellular gate formed by four S6 helix viewed from extracellular side was shown in the lower of the right panel, respectively.”.

Figure 6*. Old and revised versions of Figure 1. More labels were added in panel b of figure1, and the color of the labels was improved in the right side of panel c.

REVIEWERS' COMMENTS

Reviewer #1 (Remarks to the Author):

Authors adequately addressed all my questions.

Reviewer #2 (Remarks to the Author):

Authors addressed all my concerns and I have no more comments.

Reviewer #3 (Remarks to the Author):

The authors addressed all my comments.